# Estimation of Maize Foliar Temperature and Stomatal Conductance as Indicators of Water Stress Based on Optical and Thermal Imagery Acquired Using an Unmanned Aerial Vehicle (UAV) Platform

**Kiara Brewer [1,\*], Alistair Clulow [1], Mbulisi Sibanda [2,3], Shaeden Gokool [4], John Odindi [5], Onisimo Mutanga [5], Vivek Naiken [1], Vimbayi G. P. Chimonyo [3,6] and Tafadzwanashe Mabhaudhi [3,7]**

1   Discipline of Agrometeorology, School of Agricultural, Earth and Environmental Sciences, University of KwaZulu-Natal, Pietermaritzburg 3209, South Africa; clulowa@ukzn.ac.za (A.C.); naikenv@ukzn.ac.za (V.N.)
2   Department of Geography, Environmental Studies & Tourism, Faculty of Arts, University of the Western Cape, P Bag X17, Bellville, Cape Town 7535, South Africa; msibanda@uwc.ac.za
3   Centre for Transformative Agricultural and Food Systems, School of Agricultural, Earth and Environmental Sciences, University of KwaZulu-Natal (UKZN), Pietermaritzburg 3209, South Africa; chimonyov@ukzn.ac.za (V.G.P.C.); mabhaudhi@ukzn.ac.za (T.M.)
4   Centre for Water Resources Research, School of Agricultural, Earth and Environmental Science, University of KwaZulu-Natal, Pietermaritzburg 3209, South Africa; gokools@ukzn.ac.za
5   Discipline of Geography and Environmental Science, School of Agricultural Earth and Environmental Sciences, University of KwaZulu-Natal, Pietermaritzburg 3209, South Africa; odindi@ukzn.ac.za (J.O.); mutangao@ukzn.ac.za (O.M.)
6   International Maize and Wheat Improvement Center (CIMMYT)-Zimbabwe, Mt Pleasant, Harare P.O. Box MP 163, Zimbabwe
7   International Water Management Institute (IWMI), Pretoria 0184, South Africa
\*   Correspondence: 216012945@stu.ukzn.ac.za

**Abstract:** Climatic variability and extreme weather events impact agricultural production, especially in sub-Saharan smallholder cropping systems, which are commonly rainfed. Hence, the development of early warning systems regarding moisture availability can facilitate planning, mitigate losses and optimise yields through moisture augmentation. Precision agricultural practices, facilitated by unmanned aerial vehicles (UAVs) with very high-resolution cameras, are useful for monitoring farm-scale dynamics at near-real-time and have become an important agricultural management tool. Considering these developments, we evaluated the utility of optical and thermal infrared UAV imagery, in combination with a random forest machine-learning algorithm, to estimate the maize foliar temperature and stomatal conductance as indicators of potential crop water stress and moisture content over the entire phenological cycle. The results illustrated that the thermal infrared waveband was the most influential variable during vegetative growth stages, whereas the red-edge and near-infrared derived vegetation indices were fundamental during the reproductive growth stages for both temperature and stomatal conductance. The results also suggested mild water stress during vegetative growth stages and after a hailstorm during the mid-reproductive stage. Furthermore, the random forest model optimally estimated the maize crop temperature and stomatal conductance over the various phenological stages. Specifically, maize foliar temperature was best predicted during the mid-vegetative growth stage and stomatal conductance was best predicted during the early reproductive growth stage. Resultant maps of the modelled maize growth stages captured the spatial heterogeneity of maize foliar temperature and stomatal conductance within the maize field. Overall, the findings of the study demonstrated that the use of UAV optical and thermal imagery, in concert with prediction-based machine learning, is a useful tool, available to smallholder farmers to help them make informed management decisions that include the optimal implementation of irrigation schedules.

**Keywords:** drones; foliar temperature; machine learning; maize phenotyping; precision agriculture; smallholder farming systems; stomatal conductance; thermal imagery; UAV applications

## 1. Introduction

In recent decades, agricultural production in sub-Saharan Africa has been threatened by water scarcity, unpredictable weather, and arid conditions [1,2]. In South Africa, smallholder agriculture (less than two hectares in size) is predominantly rainfed, which often results in crops experiencing water stress and moisture shortages due to inadequate rainfall [3–5]. However, there are limited, spatially explicit, evidence-based frameworks and instruments for monitoring crop water stress in smallholder croplands, especially in those cultivating maize (*Zea mays* L.), predominantly for subsistence [6,7]. Since maize is a staple grain crop and one of the most widely cultivated crops in South African smallholder farms [8], there is a need for spatially explicit methods to characterise maize water stress to prevent yield losses and optimise the productivity of smallholder farmers.

Maize requires between 450 and 600 mm of water per season as it is sensitive to water stress, especially during the tasseling, silking, and pollination stages [9]. At physiological maturity, a single maize crop requires approximately 250 L of water to produce approximately 15 kg of grain for each millimeter of water consumed [10]. Although additional factors such as soil nutrients, light and humidity may affect growth, water stress is often the major limiting factor. The high variability of rainfall in South Africa often results in there being less water to sustain optimal crop growth and productivity [11,12]. These water deficits result in the stomata cells, which are found in the epidermis of the leaf, closing to reduce moisture loss through transpiration, resulting in increased leaf temperatures due to the limited moisture conductance available to cool the leaf surface [13,14]. Considering that maize water use varies in the emergence, vegetative and reproductive phenological stages, its phenological metrics will also vary with the variation in these growth stages along with its attributes, such as temperature and stomatal conductance, associated with the respective photosynthetic activities. Hence, the determination of foliar temperature and stomatal conductance are often used as proxies for the near-real-time detection of crop water stress [15–19]. The accurate quantification of maize foliar temperature and stomatal conductance across the growing season can, therefore, assist smallholder farmers in adopting measures to mitigate losses and optimise yield.

Traditionally, crop water stress has been determined using in situ plant measurements, soil moisture content or meteorological variables [17]. However, these approaches are time-consuming, costly, laborious, [15], and, in South Africa, prone to vandalism and theft, and thus not suitable for the continuous and real-time monitoring of crop water stress. Recently, studies have demonstrated that remote sensing techniques using optical or thermal imagery can be used to monitor crop water stress [20–24]. According to El-Hendawy, Al-Suhaibani [25], several multispectral regions of the electromagnetic spectrum are indirect water stress indicators and useful for quantifying crop water content through the leaf's biochemical attributes [25]. Specifically, the visible (blue, green, red) and the near-infrared (NIR) wavelengths hold great potential for the prediction of water due to their absorption of water through leaf pigments such as chlorophyll [26]. Moreover, the thermal infrared portion of the electromagnetic spectrum is directly correlated with water content proxies, such as temperature and stomatal activity, and is thus proficient in the analysis of temperature attributes. The thermal infrared radiation ranges from 8 μm to 14 μm on the electromagnetic spectrum, and its utility in remote sensing enables the detection of water stress due to its non-destructive nature and low labour inputs [18].

While traditional, satellite-based remote sensing techniques have proven useful in quantifying water stress, several constraints limit their suitability for monitoring temperature and stomatal conductance at the farm-scale. The spatial resolution of satellite earth observation data is generally too coarse to capture the spatial heterogeneity within smallholder farms. Moreover, broad-band thermal satellite imagery results in geometrical inaccuracies when co-registering to other portions of the electromagnetic spectrum that have higher spatial resolutions [27]. Thus, alternate approaches that can adequately capture the spatial heterogeneity at localised levels are required to facilitate precision agricultural applications within smallholder farms.

In recent years, unmanned aerial vehicles (UAVs) have become a popular field phenotyping platform for precision agricultural applications. UAV-based phenotyping is the measurement and analysis of plant characteristics in a far more advanced and detailed manner [23,24,28–35]. UAVs mounted with very-high-resolution (VHR) cameras offer advanced crop image throughput analytics and are effective in overcoming the limitations of satellite imagery [36,37]. Phenological metrics can be derived using the spatially explicit UAV images acquired by VHR cameras that offer near-real-time spectral information, which is useful for detecting gradual changes in crop phenology, foliar temperature, stomatal conductance and moisture content [23,38]. UAV images can be continuously acquired under user-defined ground sampling distances and temporal intervals, which limit atmospheric perturbations such as cloud cover [36,39]. Therefore, the accurate mapping and analysis of agricultural maize fields using an optical and thermal infrared UAV holds significant potential for providing data that inform smallholder farmers on potential crop water stress.

Foliar temperature and stomatal conductance can be optimally assessed using a robust machine learning algorithm that can derive a relationship using spectral bands and vegetation indices (VIs) to predict temperature and conductance. The use of machine learning algorithms such as random forest, support vector machines and multiple linear regression has proven to be instrumental in characterising crop characteristics such as the water and health status of plants [40–43]. The random forest ensemble is often preferred as it has generally been shown to outperform the other two algorithms [44,45]. Random forest is nonparametric algorithm that is renowned for its ability to discern the minute variations in numerous spectral variables while circumventing the impact of collinearity, autocorrelation of variables and overfitting the models. Additionally, RF has the advantage of the fact that, regardless of the sample size, it employs a bootstrapping mechanism that optimises the resampling and utilisation of the data implemented when drawing training datapoints for building trees for each model [46]. In this regard, RF was anticipated to be a suitable technique for accurately estimating foliar temperature and stomatal conductance as indicators of crop water stress in smallholder farms. VIs are mathematical combinations of image bands, which are ratioed for the extraction of spectral properties, such as canopy cover, vigour and growth [47,48]. VIs, such as the normalised difference water index (NDWI) and the normalised difference vegetation index (NDVI), have been identified as being particularly useful in directly or indirectly quantifying water stress within vegetation [13,25,49,50]. Furthermore, the use of these Vis, in combination with machine learning algorithms, have proven to be useful in characterising crop temperature and water stress [50,51]. Considering the potential to utilise UAV-derived data combined with a machine learning algorithm, in this study, we aimed to explore the utility of using UAV image data to predict foliar temperature and stomatal conductance. Specifically, we sought to predict maize foliar temperature and stomatal conductance using UAV-derived spectral variables (bands and VIs) to quantify potential water stress throughout the growing season within a smallholder farm. Furthermore, to understand the crop water stress component of the research, the phenological metrics of canopy temperature and its difference to air temperature (Tc–Ta) were plotted over phenotyping.

## 2. Materials and Methods

### 2.1. Study Site Description

Data for this study were collected over a four-month period (from February 2021 to May 2021) in the communal area of Swayimani, KwaZulu-Natal, South Africa (29°31′24″ S; 30°41′37″ E) (Figure 1), which lies within the uMshwathi Local Municipality. Swayimani is a small town of approximately 36 km$^2$ and is situated in a north-easterly direction from Pietermaritzburg. The prime activity that locals of Swayimani partake in is semi-subsistence farming on their housing plots, which is a crucial form of livelihood sustenance and food security. The predominant crop types in Swayimani are white and yellow maize, sugarcane, tomatoes, amadumbe (taro), and sweet potato [28]. The smallholder farmers manually sow crop seeds and fertilise croplands with livestock manure. Maintenance of farm plots

is purely manual, i.e., fields are hand-weeded or controlled using backpack herbicide sprayers. Crop yields are also harvested by hand. This study examined a $30 \times 96$ ($2850$ m$^2$) smallholder maize field that was situated on a gradual slope, with a field elevation ranging from 850 m to 839 m.

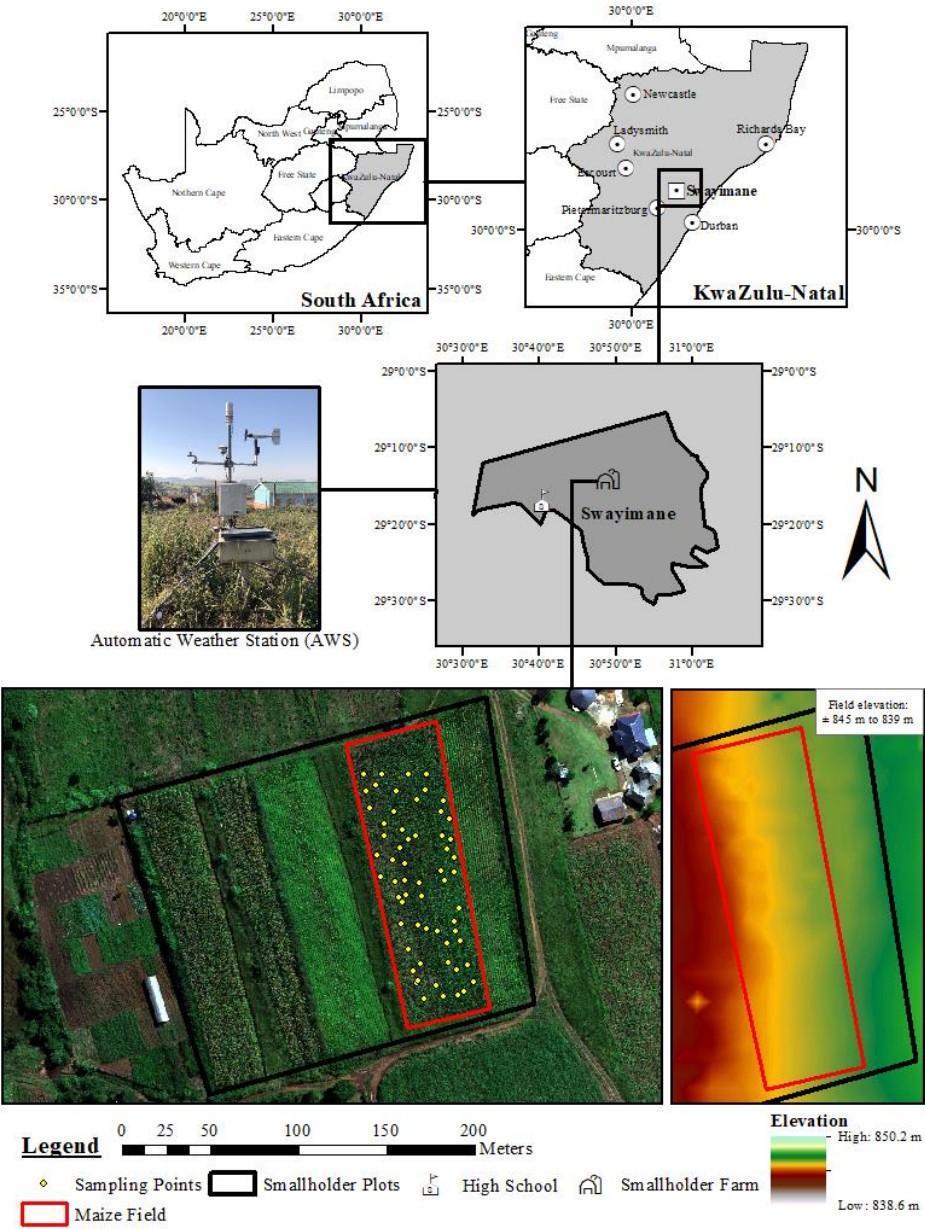

**Figure 1.** The location of the Swayimani study area, AWS, study site, and the specific smallholder maize field (WGS84, UTM zone 36S).

Agriculture and crop production in Swayimani is supported by the warm wet summers and cool dry winters, which have average temperatures ranging between 12 °C and 24 °C. The average annual rainfall ranges between 600 and 1200 mm, with most of the rain occurring during the summer, occasioned by thunderstorms. Hence, the primary source of cropland irrigation is rainwater. During data collection, Swayimani had an average daily maximum air temperature of 24 °C and total rainfall of 242.80 mm, amongst additional weather data (Figure 2). Weather conditions were recorded continuously by the Automatic Weather Station (AWS) installed at a Swayimani high school. Since the AWS was located approximately 2 km from the smallholder maize farm, it was considered proximally adequate for recording the weather conditions of the study site.

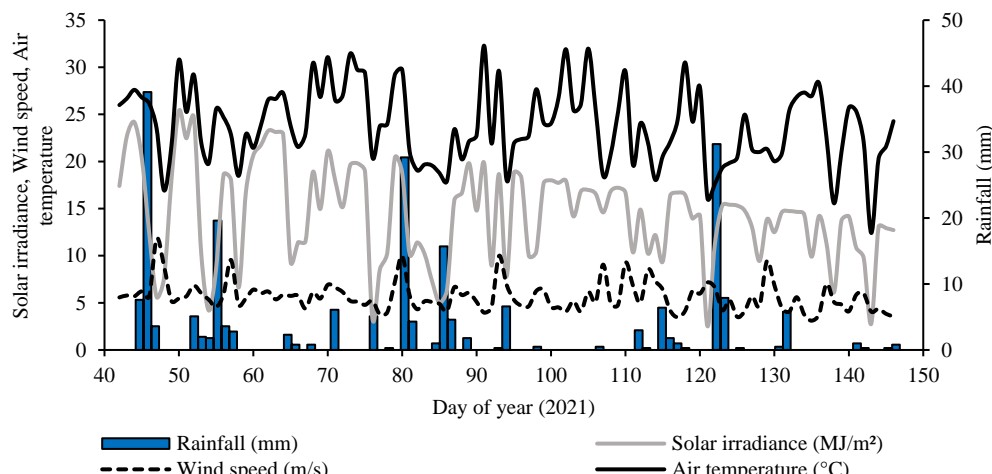

**Figure 2.** Daily weather conditions in Swayimani over the period of maize phenotyping.

### 2.2. Maize Growth Stages

The maize crop had a total growth cycle of 108 days. The seedlings were planted on 8 February 2021 and crop yields were harvested on 26 May 2021 (Table 1), Temperature and stomatal conductance measurements differed throughout the maize phenological cycle. The maize growth cycle consists of vegetative stages, where the stages from emergence to tasseling were examined (dependent on the number of fully expanded leaves), as well as the reproductive stages, which range from silking to physiological maturity (dependent on the degree of kernel development) [52,53]. Within the various stages, certain transitions are significant when monitoring the potential occurrence of water stress. These are the plant emergence stage (date of the first photosynthetic activity, phrased as VE), tasseling stage (date when the leaf area is at its largest and tassels begin to emerge, phrased as VT) and initiation of senescence (date when the green pigment of the leaf visibly begins to decrease) [10].

It is worth noting that, during the mid-vegetative stage, the western portion (lower elevation) of the field appeared to have unhealthy leaves. This may have been due to the fact that this section of cropland was not weeded with the rest of the field during the early vegetative growth stage. However, the smallholder farmer applied herbicide during the mid-vegetative growth stage to eliminate grasses and weeds that were growing between the rows. As a consequence, the herbicide lowered the health status of these crops, as the plant suffered a mild herbicide burn.

**Table 1.** Maize growth stages (Brewer et al., 2022 [28]).

| Days after Emergence | Growth Stage | | Description | Pictures |
|---|---|---|---|---|
| 0 | VE | | Germination and emergence. Planting depth 5–8 cm. | |
| 7 | V2 | | | |
| 21 | V5 | Vegetative Growth Stages | Plant population established. Growth point 20–25 mm below surface. Leaf sheath and blades. Tassel initiation. | |
| 32 | V8 | | Ear initiation and early cob development. | |
| 38 | V10 | | | |
| 44 | V12 | | Tassel at growth point begins to develop rapidly. Active growth in lateral shoots and cob development from the sixth to eighth node above surface. Brace root development. | |
| 49 | V14 | | | |
| 56 | VT | | Tasseling stage. Silks are developing. The demand for water and nutrients is high. All leaves present. Pollination 5–10 days. | |
| 63 | R1 | Reproductive Growth Stages | | |
| 70 | | | | |
| 77 | R2–R3 | | Kernel development. Silking stage. | |
| 84 | | | | |
| 91 | | | | |
| 98 | R3–R4 | | Grain filling. Nutrients are transported to cob. Sugars converted into starch. | |
| 105 | | | | |
| 112 | R5–R6 | | Physiological maturity and drying of kernels. Starch in kernels. End of mass gain. | |
| 119 | | | | |
| 160 | R+ | | Ready for harvest. Optimal moisture and nutrients. | |

### 2.3. Field Data Collection, Sampling, and Survey

Field data were collected throughout the maize phenological cycle. A 4-m meteorological tower was installed at the center of the maize field with two infrared radiometers (IRR) (Apogee SI-111, Apogee Instruments Inc., Logan, UT, USA) (Figure 3a,b). The automated IRR sensors were installed to provide a better understanding of the plant canopy temperature, which further enhanced the understanding and conclusions being made regarding crop water stress. The SI-111 IRR measures surface temperature by converting the thermal energy radiated from the surface. The SI-111 IRR obtains a spectral range from 8 μm to 14 μm, with measurement ranges from −60 °C to 110 °C, and a manufacturer accuracy of ±0.5 °C [54]. The two IRR sensors were attached at a 23 and 45° half-angle field of view (FOV), with one centered on maize and the other obtaining an azimuth view that is perpendicular to the row direction. This was to capture the large spatial extent of the field during the experimental period. We programmed the datalogger (CR1000, Campbell Scientific, Logan, UT, USA) to output average foliar canopy temperature, from 10 s measurements, at the following intervals: 5 min, 10 min, 30 min, 60 min. IRR measurements were also aggregated to acquire daily average temperature using the 10-min data. These temporal intervals were set such that short-term and long-term changes in temperature were detected over the course of the phenological cycles of maize.

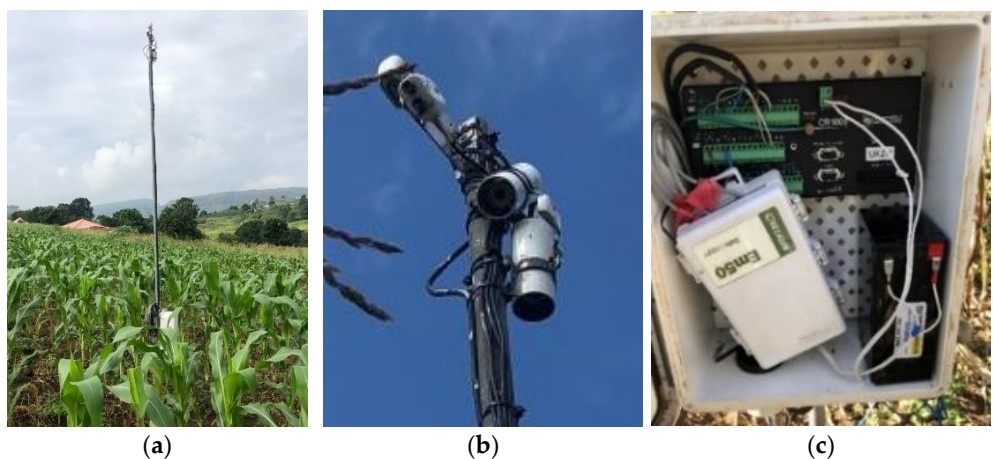

|       |       |       |
|:-----:|:-----:|:-----:|
| (**a**) | (**b**) | (**c**) |

**Figure 3.** (**a**) Automated in-field meteorological tower in the maize field, (**b**) meteorological tower mounted with SI-111 Apogee IRR sensors held 4-m above the ground, (**c**) CR1000 data logger, Em50 data logger and 12 V battery.

The SI-111 radiometers were calibrated in a temperature-controlled chamber with a built blackbody cone for the radiation source. This was carried out by holding the SI-111 IRRs in a fixture at the opening of the blackbody cone. The IRR sensors were thermally insulated from the cone and the temperature of each was independently controlled. The IRRs are held at a constant temperature while the blackbody cone was controlled at temperatures below 12 °C, above 18 °C and equal to the IRR temperature. IRR temperature data were collected at every 10 °C, until the IRRs and blackbody cone reached constant temperatures. The calibration procedure is generally conducted to standardize the measurements. IRR measurements of maize temperature were used to calibrate the handheld infrared thermometer (IRT) measurements that were used to develop the prediction model. Moreover, the IRR measurements were used to assess the thermal infrared UAV-derived temperature.

In-field maize temperature and stomatal conductance measurements were collected from the early vegetative (V5) stages to the late reproductive stages (R6), at two-week intervals. The two-week intervals were deemed adequate to capture the phenological changes in the experimental crops. The in-field measurements of the specific maize crops are collected to ground-truth the drone imagery to the actual crop characteristics on the ground. Additionally, the individual ground measurements play a crucial role in running

the prediction model. Pre-sampling of the smallholder cropland was carried out in Google Earth Pro where a polygon of the experimental field was digitised. The digitised polygon was used in ArcGIS 10.5 to produce 63 stratified random sampling points within the boundaries of the maize field. The sampling points were then imported into a handheld Trimble Global Positioning System (GPS) with sub-meter accuracy. The GPS locations were used to navigate to the maize sample points during weekly in-field data collection. Each of the maize sampling points were marked for consistent bi-weekly measurement. The 63 maize points were sampled on six occasions over phenotyping.

Specifically, a digital laser infrared GM320 handheld thermometer (IRT) was used to measure selected maize foliar temperature. The e IRT was used in this study due to its convenience and rapid ability to measure surface temperature. IRT measurements can range from approximately $-50$ °C to 330 °C. During the vegetative stages (when a sixth leaf was present) and during the tasseling stage, the IRT temperature readings were taken on the newest fully expanded leaf with an exposed collar. Subsequent to the tasseling stage, the ear leaf (i.e., the leaf attached to the same node as the primary ear shank) was evaluated [55]. Three foliar temperature measurements were taken and subsequently averaged per sampling point. Stomatal conductance was measured using a SC-1 leaf porometer (Decagon Devices, Inc., Pullman, WA, USA). The SC-1 leaf porometer calibration was used due to its ability to measure leaf conductance rates whilst taking the surrounding environmental conditions into account. Stomatal conductance is the measure of gaseous exchange (i.e., carbon dioxide intake) and transpiration (i.e., water vapour loss) through the leaf stomata, and is a function of the density, size, and degree of opening of stomata [56]. Leaves with open stomata allow for greater conductance and, consequently, indicate potentially high photosynthesis and transpiration rates, while closed leaf pores indicate potential plant stress.

The SC-1 leaf porometer calibration was performed prior to measurements under field conditions, as the leaf clip must be in thermal equilibrium with the environment. This included wetting filter paper with the distilled water provided in the sensor kit, and then placing filter paper over the hole in the calibration plate. The sensor head was then attached to the calibration plate, where a 30 s measurement began. After the measurement, the sensor was equilibrated, and the sensor head was reattached for another measurement. Calibration measurements were repeated up to 10 times until a stable measurement was achieved. Again, the calibration of the instrument is performed to standardise the measurement. Leaf porometer readings were carried out on the same leaf as the IRT temperature readings. The same readings were carried out on the same leaf, to link the actual measured leaf temperature and its stomatal conductance. During the mid and late reproductive stages, a fully matured maize leaf in full exposure to sunlight was selected, and the sensor was placed in the middle of the leaf blade perpendicular to the midrib when conducting measurements. Blades of expose leaves were used in this study to avoid the impact of shadowing, which might alter the photosynthetic activities of the plant. The SC-1 leaf porometer automatically measured the leaf stomatal conductance (in mmol m$^{-2}$ s$^{-1}$) for a measurement period of 30 s, whilst providing measurements of air temperature and humidity. Stomatal measurements close to 0 mmol m$^{-2}$ s$^{-1}$ indicate extreme stress, whereas values close to 500 mmol m$^{-2}$ s$^{-1}$ indicate no stress.

IRT (temperature) and SC-1 leaf porometer (stomatal conductance) measurements were consistently taken between 10:00 am and 2:00 pm throughout the various stages of the maize phenological cycle. These time periods were suggested due to the optimal solar radiation and limited atmospheric effects. The temperature and stomatal conductance data were then added to the 63 sampling points map in a geographical information system (GIS). The point map was overlaid with the acquired and preprocessed optical and thermal UAV imagery to derive spectral reflectance values from each sampling point.

### 2.4. UAV: DJI Matrice 300 and MicaSense Altum

The DJI Matrice 300 (DJI M-300) series platform and the MicaSense Altum camera and Downwelling Light Sensor 2 (DLS-2) were used to obtain aerial-view flights of the smallholder cropland. The DJI M-300 platform has four rotary-wings and vertical take-off and landing (VTOL) technology, which is well-suited to taking-off and landing in the rural areas in close proximity to settlements (Figure 4a). The novelties of the DJI M-300 series platform include its 15 km transmission range, 7000 m maximum flight height, obstacle detection and avoidance, flightpath planning and embedded GPS tracker. The maximum flight time of the M-300 with the mounted Altum was approximately 42 min and could reach a speed of 27 m/s, outperforming many drone platforms on the market. It is because of these novelties that the DJI-M-300 was selected and used in this study. Moreover, the MicaSense Altum camera is an optical and thermal infrared imaging camera that encompasses five spectral high-resolution narrow bands (blue, green, red, red-edge, and near-infrared) and a radiometric longwave infrared thermal sensor (Figure 4b). The Altum camera was specifically used in this research as it offers synchronised optical and thermal infrared image capture and has a global shutter that supports up to a one-second capture rate for high spatial resolution and aligned imagery [57]. The optical bands have a 2064 × 1544 at 120 m (3.2 megapixels per optical band) sensor resolution and a ground sample distance (GSD) of 5.2 cm per pixel at a height of 120 m. The flight height was chosen to suit the battery life of the platform while adequately covering the study area. The thermal infrared camera has a 160 × 120 sensor resolution and a GSD of 81cm per pixel at 120 m (Table 2). The optical camera has a 48° × 37° FOV, with an 8 mm focal length, while the thermal camera has a 57° × 44° FOV, with a 1.7 mm focal length.

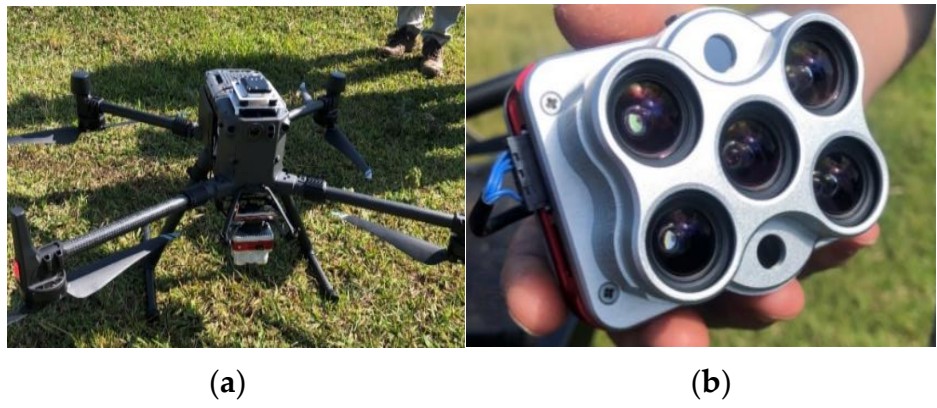

(**a**)                                   (**b**)

**Figure 4.** (**a**) DJI Matrice 300 platform and, (**b**) MicaSense Altum optical and thermal camera.

**Table 2.** MicaSense Altum camera specifications.

| Band | Spectral Color | Band Center/Range | Ground Sampling Distance at Flying Height of 120 m |
|---|---|---|---|
| 1 | Blue | 475 nm | 5.2 cm per pixel |
| 2 | Green | 560 nm | 5.2 cm per pixel |
| 3 | Red | 668 nm | 5.2 cm per pixel |
| 4 | Red-edge | 717 nm | 5.2 cm per pixel |
| 5 | Near-infrared | 842 nm | 5.2 cm per pixel |
| 6 | Thermal infrared | 8000–14,000 nm | 5.2 cm per pixel |

### 2.5. Image Acquisition and Processing

A shapefile of the Swayimani study area was created in Google Earth Pro and uploaded onto the DJI M-300 smart controller, where a flight plan was constructed over the study area (Figure 5a; Table 3). The flight plan was created in advance to enable a hands-free UAV flight over the maize cropland and adjacent areas. Before and after the flight, the camera was calibrated using the MicaSense Altum calibrated reflectance panel (CRP). The drone camera

was calibrated on every field visit to ensure the quality of the spectra. The radiometric calibration target (the CRP) is a white balance card that provides the reflectance properties of the card across the spectrum of light captured by the Altum. The process included the user holding the drone over the CRP and taking an unshaded image of the CRP to discern the light conditions of each flight date, time, and location (Figure 5b). Drone flights were conducted on a bi-weekly basis, on days with limited cloud cover. Bi-weekly monitoring was chosen for maize as it enables consistency in measurements of the crucial growth stages during vegetative and reproductive growth. Drone flights were generally conducted in the mornings, between 10:00 am and 12:00 pm, as there was optimal solar radiation. These time frames also coincided with the field sampling measurements of temperature and stomatal conductance. Field measurements were simultaneously performed with drone flight to ensure that the imagery captured the status of the canopy at that particular time of flight.

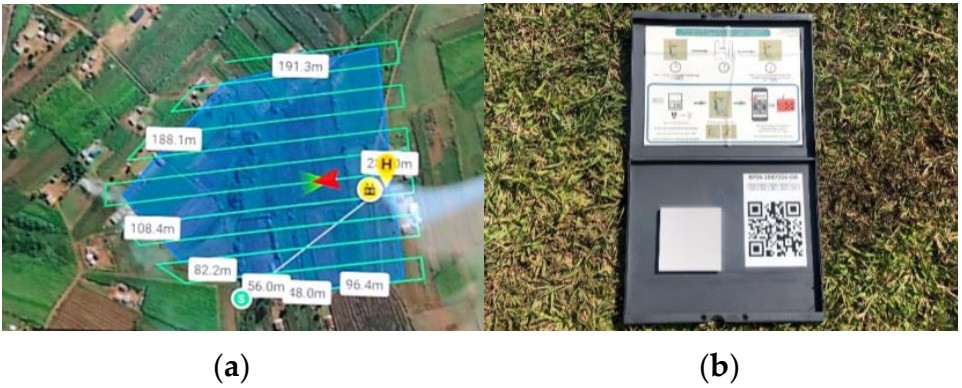

(**a**)  (**b**)

**Figure 5.** (**a**) DJI M-300 flight plan, (**b**) MicaSense Altum calibration reflectance panel.

**Table 3.** UAV flight specifications.

| Parameters | Specifications |
| --- | --- |
| Altitude | 100 m |
| Ground sampling distance (optical) | 7 cm |
| Ground sampling distance (thermal infrared) | 109 cm |
| Speed | 16 m/s |
| Flight duration | 14 min 36 s |
| Composite images | 321 |
| Image overlap | 80% |

A total of 3576 images (per flight) were processed in Pix4D photogrammetry software on a bi-weekly basis. Pix4D was used in this study due to its ability to perform stitching and atmospheric corrections of UAV-acquired data. Processing in Pix4D included image correction and mosaicking and exporting as a GeoTIFF file. Radiometric correction used all the captured images, including the CRP images that were taken before and after flight. This enabled the Pix4D radiometric correction to calibrate and correct the images reflectance to the exact atmospheric conditions during the time of image acquisition. The CRP also has an absolute reflectance, which makes it possible to compare images from different flights. Once processing was complete, an orthomosaicked image and a digital elevation model (DEM) image was produced. Georeferencing of the orthomosaic and DEM image were conducted in ArcGIS 10.5 with the use of ground reference points from Google Earth Pro and referenced to the Universal Transverse Mercator (UTM zone 36S) projection.

The LWIR thermal infrared band was converted to absolute temperature values in Pix4Dfields, using the following equation:

$$Temperature = \frac{LWIR\ Thermal\ infrared}{100} - 273.15 \tag{1}$$

The cropland reflectance data were extracted from the optical and thermal infrared bands of the drone image to derive vegetation indices and conduct a predictive analysis. The ground-truthed maize IRT temperature and stomatal conductance recorded measurements, and the GPS points of each measured maize plant (in the form of a point map) were overlaid with the UAV optical and thermal image. Reflectance data were extracted for each GPS point, and for the optical-thermal bands. The drone image data were then used to calculate the vegetation indices (Vis) presented in Table 4. The Vis were selected to assess for water-stress, including direct and indirect water-related indices, which created in Microsoft Excel. These VIs were selected based on their optimal performance in the research literature [13,58,59].

### 2.6. Statistical Analysis

The sampled data were randomly partitioned into training (70%) and testing (30%) datasets, which were used to develop the predictive regression models. A random forest regression algorithm was used to predict maize foliar temperature and stomatal conductance (from optical-thermal bands and VIs), since it is well-known for its robustness, simplicity, and ability to predict at high accuracies no matter the inputted sample size [60,61]. Random forest (RF) is a machine learning algorithm (MLA) that uses bootstrapping aggregation, which forms a number of trees on a subset of samples derived from the training dataset [40]. The decision trees are increased to their threshold using randomised UAV-derived optical-thermal datapoints. Each of the nodes were partitioned using a randomised sets of input variables [62]. Additionally, the RF MLA has the ability to identify the optical-thermal bands and VIs that were most influential in the prediction model based on the sum of the reduction in Gini impurity over the feature nodes [63]. These are referred to as the variables of importance, as they significantly contribute to the accuracy of the model output.

The RStudio software version 1.4.1564 (accessed from https://www.rstudio.com/ (accessed on the 24 February 2020) was used to produce the regression models using the RGtk2 and rattle packages. The variable importance score was used to optimise the outputs of the RF model. Any variables of low significance to the model were removed during data analysis and the RF model was continuously altered for optimal prediction. The variable selection process limits anomalies in multicollinearity and redundancy, which, in turn, have implications for the regression models performance. The model was optimised and fine-tuned by the user to hyper-parameters of 500 trees and 6 variables for temperature, and 500 trees and 10 variables for stomatal conductance. These hyper-parameters were attained after numerous iterations.

**Table 4.** UAV derived optical-thermal vegetation indices.

| Vegetation Index | Abbreviation | Equation | Reference |
|---|---|---|---|
| Direct water-related indices | | | |
| Normalised difference water index | NDWI | $\frac{GREEN-NIR}{GREEN+NIR}$ | Yang and Du [59], Gao [64] |
| Indirect water-related indices | | | |
| Normalised difference vegetation index | NDVI | $\frac{NIR-RED}{NIR+RED}$ | Panigrahi and Das [58] |
| Transformed normalised difference vegetation index | TDVI | $\sqrt{\frac{NIR-RED}{NIR+RED}+0.5}$ | Castellanos-Quiroz, Ramírez-Daza [65] |
| Normalised difference red edge index | NDRE | $\frac{NIR-RED\ EDGE}{NIR+RED\ EDGE}$ | Song, Birch [66] |
| Normalised green–red difference index | NGRDI | $\frac{GREEN-RED}{GREEN+RED}$ | Song, Birch [66] |
| Green chlorophyll index | CIgreen | $\left(\frac{NIR}{GREEN}\right)-1$ | Zhang and Zhou [13] |
| Red-edge chlorophyll index | CIrededge | $\left(\frac{NIR}{RED\ EDGE}\right)-1$ | Zhang and Zhou [13] |
| Green NDVI | GNDVI | $\frac{NIR-GREEN}{NIR+GREEN}$ | Song, Birch [66] |

**Table 4.** *Cont.*

| Vegetation Index | Abbreviation | Equation | Reference |
|---|---|---|---|
| Canopy chlorophyll content index | CCCI | $\frac{\mathrm{NDRE}-\ \mathrm{NDRE_{min}}}{\mathrm{NDRE_{max}}-\ \mathrm{NDRE_{min}}}$ | Fitzgerald, Rodriguez [67] |
| Chlorophyll vegetation index | CVI | $\mathrm{NIR} \times \left( \frac{\mathrm{RED}}{\mathrm{GREEN^2}} \right)$ | Vincini and Frazzi [68] |
| Enhanced vegetation index | EVI | $\frac{2.5(\mathrm{NIR}-\mathrm{RED})}{\mathrm{NIR}+6\mathrm{RED}-7.5\mathrm{BLUE}+1}$ | Wiratmoko, Prasetyo [69] |
| Soil adjusted vegetation index | SAVI | $\frac{(\mathrm{NIR}-\mathrm{RED})(1+\mathrm{L})}{\mathrm{NIR}+\mathrm{RED}+\mathrm{L}}$ | Sishodia, Ray [70] |
| Optimised soil-adjusted vegetation index | OSAVI | $\frac{1.16\,(\mathrm{NIR}-\mathrm{RED})}{\mathrm{NIR}+\mathrm{RED}+0.16}$ | Sishodia, Ray [70] |

*2.7. Accuracy Assessment*

The predicted regression models of foliar temperature and stomatal conductance were tested for their performance accuracy using root–mean-squared error (RMSE), coefficient of determination ($R^2$), and the relative root–mean-squared error (RRMSE). The RMSE tests the degree of error found between the in-field measurements and predicted outputs of foliar temperature and stomatal conductance; the $R^2$ measures the variance between the in-field measured and predicted maize lead foliar temperature and stomatal conductance; the RRMSE is used to compare the performance of prediction models over the maize growth cycle. To derive the RRMSE, the RMSEs of the prediction models were normalised using the average of each variable. This is then expressed as a percentage, where low percentages are more accurate [71]. Accuracy assessments were conducted in Microsoft Excel.

## 3. Results

*3.1. Descriptive Analysis of UAV-Derived Data and SI-111 IRR Maize Temperature Data*

3.1.1. Maize Temperature Data over Phenotyping

The IRR time-series data were used to plot the difference between IRR foliar canopy temperature (Tc) and air temperature (Ta) (Figure 6). The foliar-canopy-to-air-temperature difference (Tc–Ta) fluctuated throughout maize phenotyping, as foliar canopy temperatures were influenced by ambient conditions of air temperature, solar radiation, and the influence of rainfall. Hence, a similar fluctuation trend was followed for solar radiation and Tc–Ta. Days of low solar radiation or rainfall were associated with a lower Tc–Ta, and days of high solar radiation were generally associated with a higher Tc–Ta.

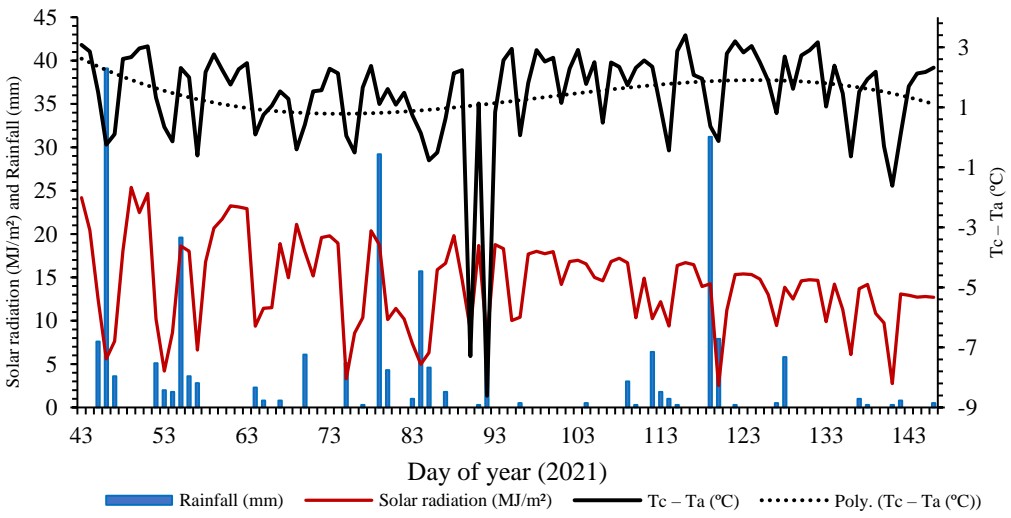

**Figure 6.** Daily average Tc–Ta, solar radiation, and rainfall throughout the maize phenological cycle.

The trendline through Tc–Ta over the maize growth season shows that a higher Tc–Ta was associated with early vegetative stages, such as DOY 43 at 3.1 °C, as well as mid-

reproductive and late reproductive growth stages, such as DOY 115 at 3 °C and DOY 130 at 2.7 °C, respectively. The maximum Tc–Ta was recorded during the mid-reproductive stage on DOY 116 at 3.4 °C. This was subsequent to a hailstorm that occurred on DOY 113, which increased the Tc–Ta, and also resulted in Tc–Ta remaining relatively high for the duration of phenotyping at approximately 1.9 °C. A lower Tc–Ta was associated with the mid-vegetative and late vegetative growth stages, such as DOY 85 at −0.8 °C and DOY 90 at −7.3 °C, respectively. The lowest Tc–Ta was recorded during the late vegetative stage on DOY 92 at −8.6 °C.

Generally, the solar radiation and air temperature decreased as the winter season approached; however, the Tc–Ta increased as the winter season approached, suggesting reduced transpiration as a result of water stress during the mid-reproductive and senescence in the late reproductive growth stages.

### 3.1.2. Evaluation of UAV Thermal Imagery against In-Field SI-111 IRR Temperature Sensors

The thermal UAV-derived temperature had a strong positive linear relationship ($R^2 = 0.94$; $p = 0.001$) with the in-field IRR temperature sensors over the various maize growth stages (Figure 7) for the days on which we collected UAV imagery. During the early vegetative growth stage, DOY 61, the IRR and UAV temperatures recorded a satisfactory correlation of approximately 29.7 °C to 30.8 °C, respectively. The mid-vegetative stage, DOY 77, obtained a significant correlation of 22.2 °C and 21.2 °C for IRR and UAV temperatures.

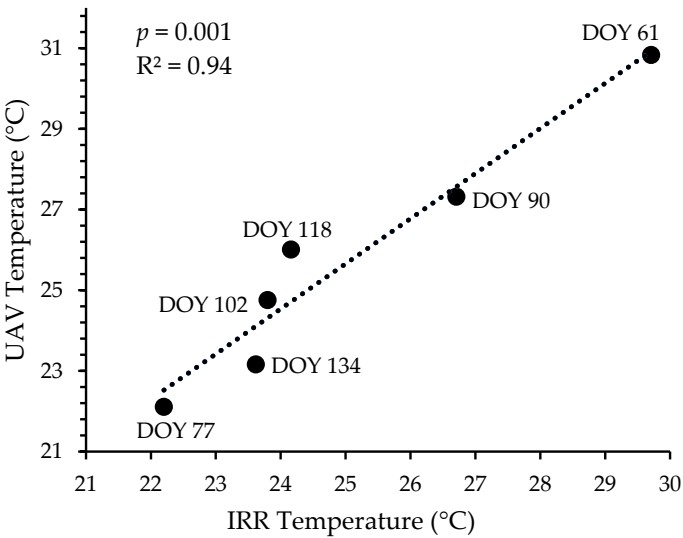

**Figure 7.** Correlation of in-field IRR sensors and UAV-derived temperature throughout the maize phenological cycle.

Moreover, during the late vegetative stage (DOY 90) and the early reproductive stage (DOY 102), maize IRR and UAV temperatures substantially correlated and were recorded at approximately 27 °C and 24 °C, respectively. However, the correlation during the mid-reproductive and late reproductive stages, DOY 118 and DOY 134, deviated from the trendline, due to the maize canopy disturbance caused by the hailstorm on DOY 113. Hence, IRR and UAV temperatures were recorded at 24 °C and 26 °C for the mid-reproductive stage, and 24 °C and 23 °C for late reproductive stage.

### 3.2. Descriptive Statistics of In Situ Maize Temperature and Stomatal Conductance Measurements

The IRR and IRT minimum and the maximum temperatures of the 63-sampling points were used to illustrate the descriptive temperature statistics (Table 5). The average maximum IRR temperature for the entire phenological cycle was 32.2 °C, and the average IRT temperature for the maize growth cycle was 32.6 °C. This suggested an average

temperature offset of 0.4 °C between the IRR and IRT temperature measurements. The highest recorded IRR temperature occurred during the late vegetative growth stage of V14–VT at 35 °C, and the maximum IRT temperature was 39.1 °C, recorded during the V14–VT growth stage. The average minimum IRR temperature for the entire phenological cycle was 18.3 °C, and the average IRT temperature for the maize growth cycle was 20.9 °C. This suggested an average temperature offset of 2.6 °C between the IRR and IRT temperature measurements.

**Table 5.** Descriptive statistics of IRR and IRT foliar temperature throughout the maize growth stages.

| Maize Foliar Temperature at the Various Growth Stages | | Maximum (°C) | | Minimum (°C) | |
|---|---|---|---|---|---|
| | | IRR | IRT | IRR | IRT |
| DOY 61 | V5–V10 | 34 | 32.7 | 17 | 21.5 |
| DOY 77 | V12 | 33 | 23.4 | 20.5 | 15.5 |
| DOY 90 | V14–VT | 35 | 39.1 | 17.1 | 23.1 |
| DOY 102 | R1–R3 | 34 | 33.7 | 18.9 | 21.4 |
| DOY 118 | R3–R4 | 33 | 34.3 | 16 | 19.3 |
| DOY 134 | R5–R6 | 30 | 32.3 | 20.4 | 24.8 |
| | Mean | 33.2 | 32.6 | 18.3 | 20.9 |
| | Median | 33.5 | 33.2 | 18 | 21.5 |
| | Standard deviation | 1.7 | 5.1 | 1.9 | 3.2 |
| | Co-efficient of variation | 5.2 | 15.7 | 10.4 | 15.5 |

The lowest IRR temperature value occurred during the R3–R4 growth stage at 16 °C, whereas the lowest IRT temperature value occurred during the V12 growth stage at 15.5 °C. The IRR maximum values were within 1.7 standard deviations of the mean, whereas the IRT maximum values were within 5.1 standard deviations of the mean. Similarly, with the minimum values, the standard deviation of the IRR and IRT temperatures were 1.9 and 3.2, respectively. The total mean coefficient of variation for the maximum IRR was 5.2%, whereas the maximum coefficient of variation for the IRT was 15.7%. Moreover, the minimum IRR and IRT temperatures had coefficients of variation of 10.4% and 15.5%, respectively. The IRR values suggest precise temperature estimates that are close to the mean value.

The measured maize stomatal conductance varied over different stages of maize phenotyping (Table 6). The average stomatal conductance over the maize phenotyping was 206.9 mmol m$^{-2}$ s$^{-1}$. The lowest conductance value occurred during the early vegetative growth stage (V5–V10) at 42 mmol m$^{-2}$ s$^{-1}$, and the highest stomatal conductance occurred during the early reproductive development stage (R1–R3) at 556.5 mmol m$^{-2}$ s$^{-1}$. Reproductive stages were characterised by higher stomatal conductance values compared to the vegetative stages. However, the average stomatal conductance for the mid-reproductive stage (R2–R4) decreased to 172.6 mmol m$^{-2}$ s$^{-1}$ due to crop stress from the hailstorm that occurred on DOY 113. Furthermore, the median value of maize stomatal conductance across the growing season was 194.6 mmol m$^{-2}$ s$^{-1}$, and the average stomatal conductance values were within 79.3 standard deviations of the mean value.

Importantly, foliar temperature and stomatal conductance had a significant inverse relationship, producing an R$^2$ = 0.72 (Figure 8). The negative relationship of stomatal conductance and temperature further enhanced the identification of potential crop water stress. This relationship illustrated that, when stomatal conductance was low, foliar temperatures were high, i.e., a hot canopy. Furthermore, high stomatal conductance was associated with low foliar temperatures, i.e., a cool canopy, suggesting optimal maize water productivity.

**Table 6.** Descriptive statistics of stomatal conductance throughout the maize phenological cycle.

| Maize Stomatal Conductance at the Various Growth Stages | | Minimum (mmol m$^{-2}$ s$^{-1}$) | Maximum (mmol m$^{-2}$ s$^{-1}$) | Mean (mmol m$^{-2}$ s$^{-1}$) | Median (mmol m$^{-2}$ s$^{-1}$) | Standard Deviation |
|---|---|---|---|---|---|---|
| DOY 61 | V5–V10 | 42 | 245.1 | 121.8 | 112.9 | 49.25 |
| DOY 77 | V12 | 86.6 | 556.5 | 248.5 | 238.1 | 113.3 |
| DOY 90 | V14–VT | 44.2 | 404.8 | 166.5 | 157.6 | 73.7 |
| DOY 102 | R1–R2 | 182.7 | 480.1 | 298.9 | 290.1 | 79.3 |
| DOY 118 | R2–R4 | 100.2 | 373.6 | 172.6 | 160.3 | 55.6 |
| DOY 134 | R4–R5 | 74.3 | 483.1 | 233.3 | 208.5 | 104.8 |
| **Average value** | | 88.3 | 423.9 | 206.9 | 194.6 | 79.3 |

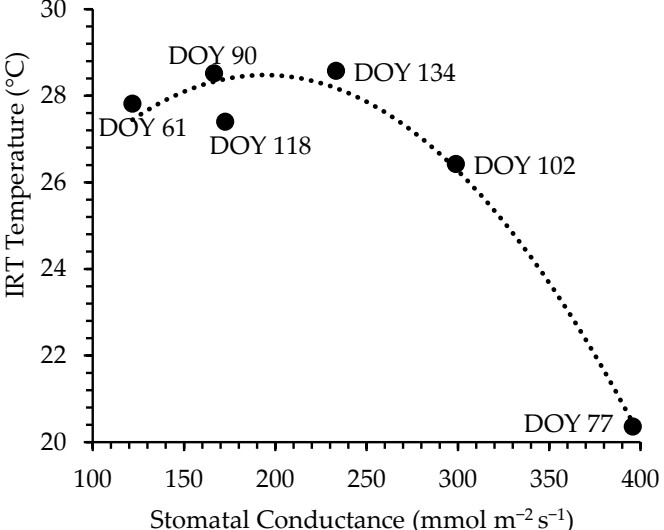

**Figure 8.** Correlation between foliar temperature and stomatal conductance throughout the maize phenological cycle.

For example, on DOY 77, the maize temperature was low, at 20 °C, and the stomatal conductance was high, at 396 mmol m$^{-2}$ s$^{-1}$, illustrating potentially optimal crop conditions. On DOY 61, the maize temperature was higher, at 28 °C, and the stomatal conductance was low, at 122 mmol m$^{-2}$ s$^{-1}$, indicating potential water stress. Thus, the inverse relationship between foliar temperature and stomatal conductance were useful for estimating crop water stress.

### *3.3. UAV-Derived Data: Estimation of Maize Temperature and Stomatal Conductance*

3.3.1. Optimised Regression Models of Maize Foliar Temperature and Stomatal Conductance over the Phenological Stages

For the prediction of maize temperature, the mid-vegetative stage (DOY 77 (V12)) yielded the most optimal modelled RMSE = 0.59 °C and $R^2$ = 0.81 (RRMSE = 2.9%) (Figure 9b1). The optimal variables for this model were the thermal infrared, followed by red, NGRDI, CVI and NDVI, in order of importance (Figure 10b). The mid-reproductive stages yielded an RMSE = 1.24 °C, $R^2$ = 0.76 and the poorest maize phenology RRMSE = 6.2%. Model prediction accuracies moderately improved in the late vegetative stages (DOY 90 (V14–VT)) and the early reproductive stage (DOY 102 (R1–R3)) to an RMSE = 1.14 °C, $R^2$ = 0.79 (RRMSE = 4%) and RMSE = 1.02 °C, $R^2$ = 0.73 (RRMSE = 3.9%), respectively. The model from the late reproductive stage (DOY 134 (R5–R6)) obtained an RMSE = 0.7 °C and $R^2$ = 0.78 (RRMSE = 2.6%) based on NDRE, OSAVI, CCCI, thermal infrared, and EVI, in order of importance (Figure 9f(1)). The early vegetative stage (DOY 61 (V5–V10)) exhibited an RMSE = 1.29 and $R^2$ = 0.69 (RRMSE = 4.7%).

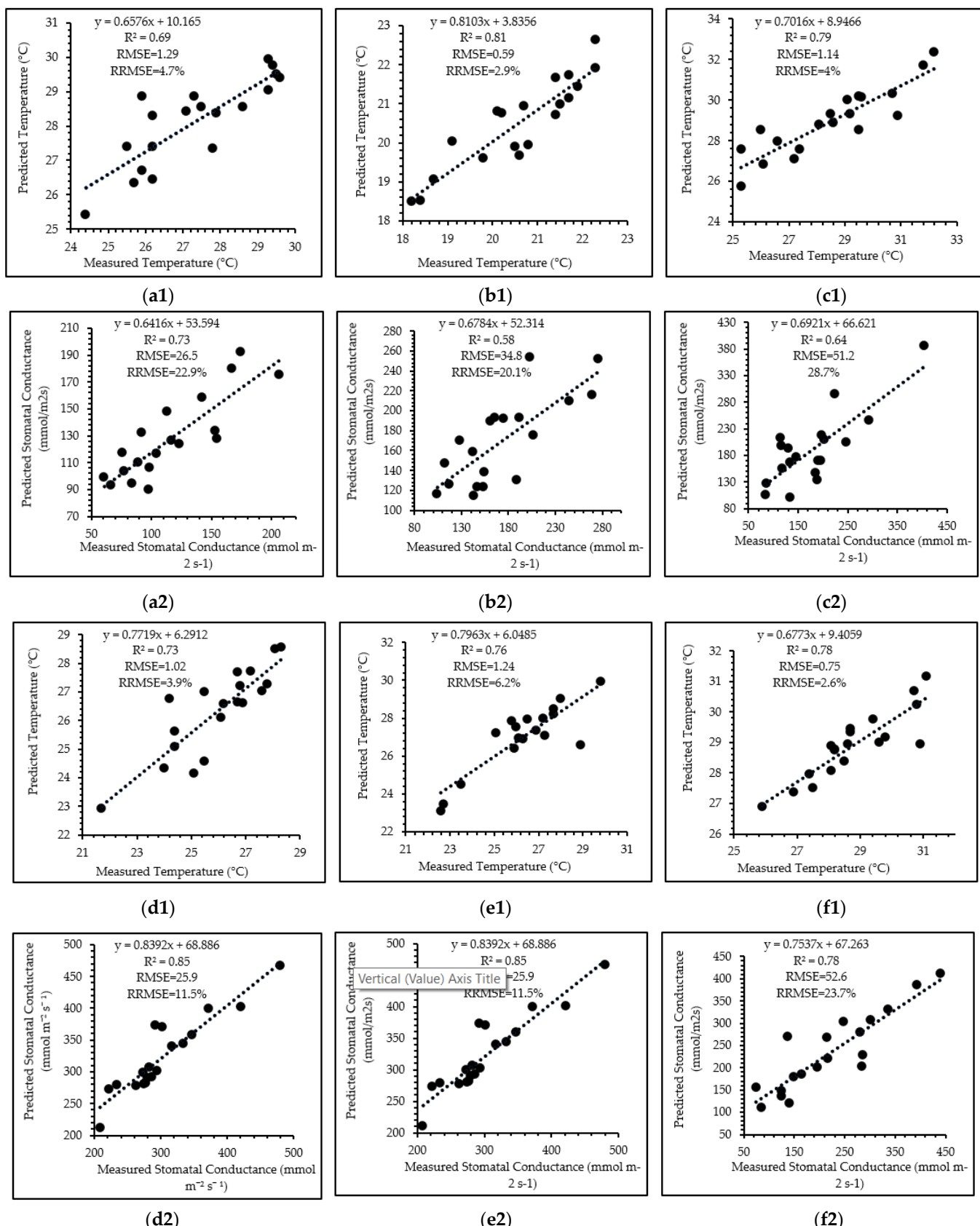

**Figure 9.** One-to-one relationships between measured and predicted IRT foliar temperature and stomatal conductance throughout the maize phenological cycle: (**a1**,**a2**) V5–V10, (**b1**,**b2**) V12, (**c1**,**c2**) V14 to VT, (**d1**,**d2**) R1–R3, (**e1**,**e2**) R3–R4, (**f1**,**f2**) R5–R6.

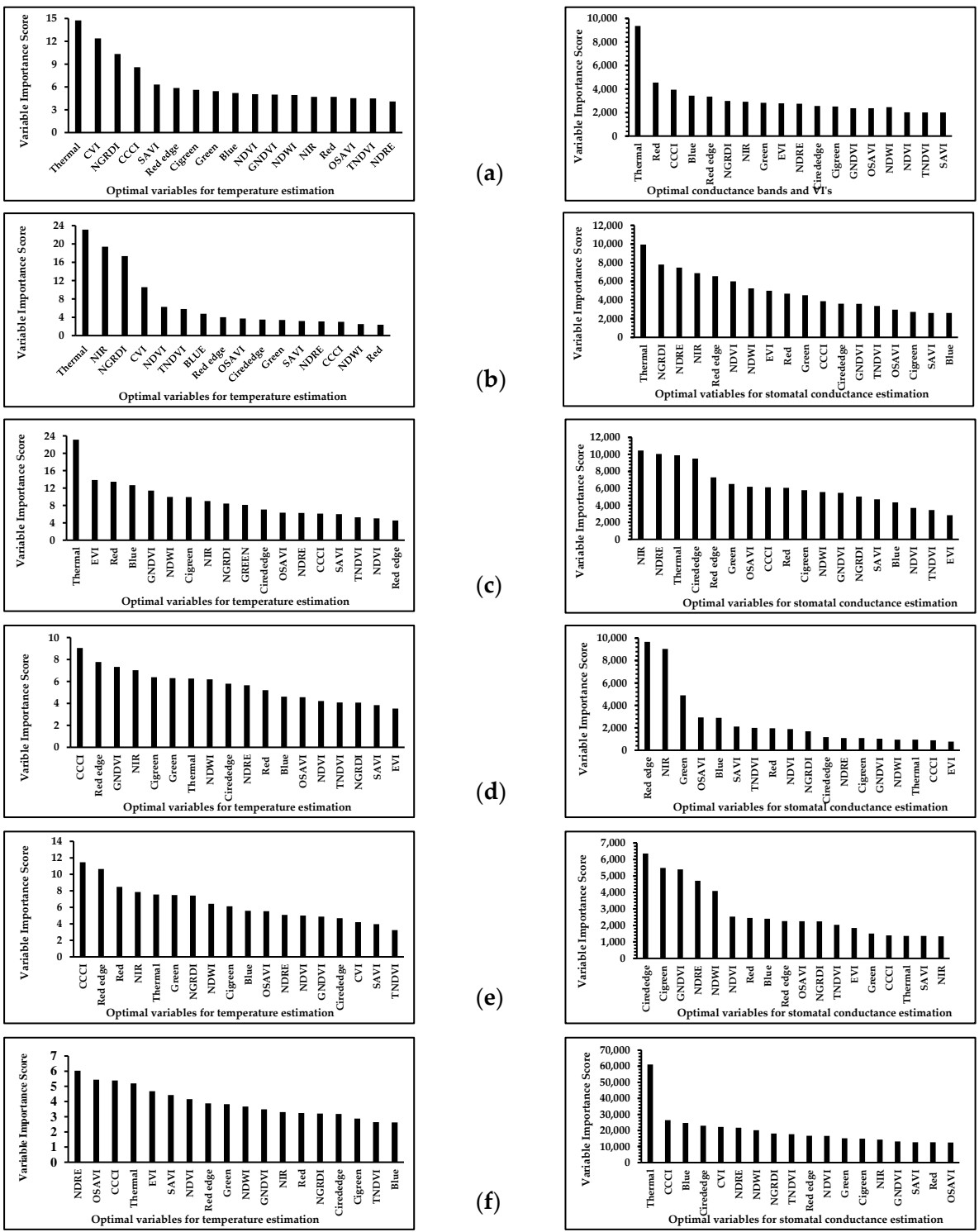

**Figure 10.** Variable importance scores of optimal foliar temperature and stomatal conductance bands and VIs. throughout the phenological cycle: (**a**) V5–V10, (**b**) V12, (**c**) V14 to VT, (**d**) R1–R3, (**e**) R3–R4, (**f**) R5–R6.

When estimating stomatal conductance, the early reproductive stage (DOY 102 (R1–R3)) produced the most accurate model, with an RMSE = 25.9 mmol m$^{-2}$ s$^{-1}$, the highest $R^2$ = 0.85 (and the best RRMSE = 11.5%) based on NIR, NDRE, Thermal, CIrededge, and red-edge, in order of importance (Figure 10d). Poor stomatal conductance accuracies were attained during the late reproductive stage (DOY 134 (R5–R6)) with an RMSE =

52.6 mmol m$^{-2}$ s$^{-1}$ and R$^2$ = 0.78 (RRMSE = 23.8%) using the thermal infrared, CCCI, blue, CIrededge, and CVI in order of importance (Figure 10f). In addition, the late vegetative stages (DOY 90 (V14–VT)) also yielded a poor model with an RMSE = 51.2 mmol m$^{-2}$ s$^{-1}$ and R$^2$ = 0.64 (poorest RRMSE = 28.7%). The mid-reproductive stage (DOY 118 (R3–R4)) model produced an RMSE = 44.6 mmol m$^{-2}$ s$^{-1}$ and R$^2$ = 0.7 (RRMSE = 25.6%). The mid-vegetative stage (DOY 77 (V12)) yielded improved model accuracies with an RMSE = 34.8 mmol m$^{-2}$ s$^{-1}$, the poorest R$^2$ = 0.58 (RRMSE = 20.1%), whereas the early vegetative stage (DOY 61 (V5–V10)) produced an optimal RMSE = 26.5 mmol m$^{-2}$ s$^{-1}$ and R$^2$ = 0.73 (RRMSE = 22.9%). The early and mid-reproductive stages were characterised by the red-edge band and the CIrededge, respectively, where the red-edge and NIR bands were a clear stand-out and optimal model contributors. However, the model achieved higher R$^2$ values during the reproductive stages and more optimal RMSE values during the vegetative stages.

3.3.2. Mapping the Spatial Distribution of Maize Temperature and Stomatal Conductance over the Various Phenological Stages

The modelled maize temperature ranged from 8 to 57 °C (Figure 11). It is evident that the maize field temperatures were high during the early vegetative growth stage. Subsequently, during the mid-vegetative growth stage, the field temperature moderately decreased, and further decreased during the late vegetative stage to reach the lowest field foliar temperatures. Likewise, in the early reproductive stage, the field was characterised by a generally low temperature, with the exception of the eastern edge (high elevation) of the field. The maize temperature during the mid-reproductive stage increased as a result of hailstorm damage. During the late reproductive stage, the hailstorm effects increased, and resulted in a further escalation of field temperatures.

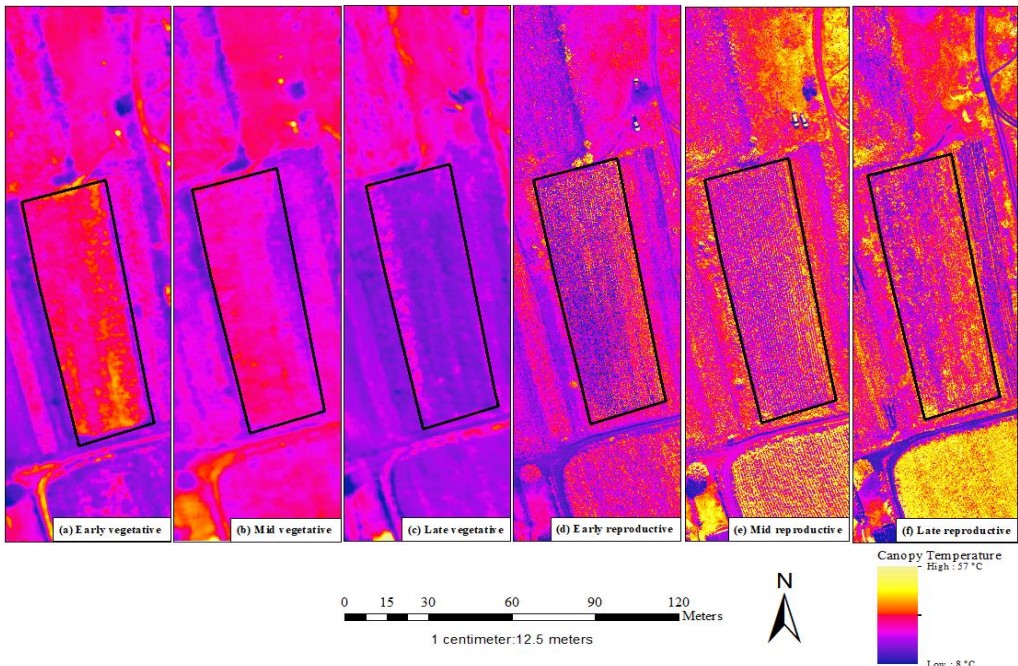

**Figure 11.** Foliar temperature of maize over the smallholder field for vegetative stages (**a**) early vegetative V5–V10, (**b**) mid vegetative V12, (**c**) late vegetative V14 to VT, and reproductive stages (**d**) early reproductive R1–R3, (**e**) mid reproductive R3–R4, (**f**) late reproductive R5–R6 (WGS84, UTM zone 36S).

The spatial distribution of stomatal conductance was estimated based on the optimal models for each maize phenological stage. The stomatal conductance values ranged from 82.2 mmol m$^{-2}$ s$^{-1}$ to 683.4 mmol m$^{-2}$ s$^{-1}$ (Figure 12). It can be observed that the stomatal

conductance of maize was relatively low throughout the maize fields. However, high levels of stomatal conductance were identified during the early vegetative stage towards the southern portion of the field, the eastern part of the field during the late vegetative stage, and the eastern section during the mid-reproductive stage. The remainder of the stages, the mid-vegetative, early reproductive, and late reproductive, were characterised by lower levels of stomatal conductance. The late reproductive stage had the lowest conductance due to the hailstorm stress and crop senescence.

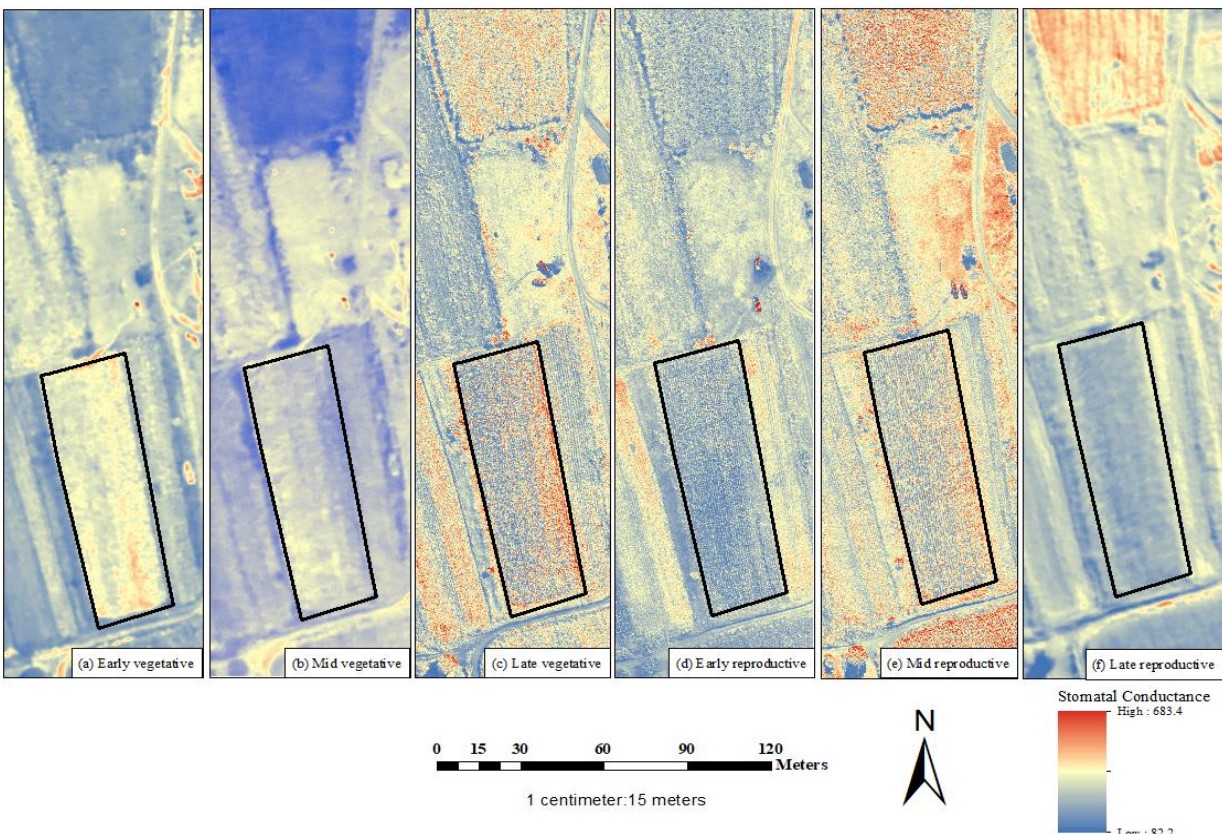

**Figure 12.** Maize stomatal conductance over the smallholder field for vegetative stages (**a**) early vegetative V5–V10, (**b**) mid vegetative V12, (**c**) late vegetative V14 to VT, and reproductive stages (**d**) early reproductive R1–R3, (**e**) mid reproductive R3–R4, (**f**) late reproductive R5–R6 (WGS84, UTN zone 36S).

## 4. Discussion

The objective of this study was to predict maize temperature and stomatal conductance over the various maize growth stages using UAV-derived data in combination with the random forest algorithm. We aimed to determine the most optimal maize growth stage(s) for temperature and stomatal conductance model estimation. It is evident that maize foliar temperatures and stomatal conductance differed throughout phenotyping, and the UAV-derived data could discern the optimal growth stages for the characterization of temperature and stomatal conductance as proxies for crop water stress. For this purpose, the foliar temperature and stomatal conductance data were used to understand the potential crop water stress and moisture status of the smallholder field throughout the growing stages.

### 4.1. Prediction of Maize Water Stress Using Foliar Temperature and Stomatal Conductance

The regression models were set to predict maize foliar temperature and stomatal conductance using the thermal infrared and optical UAV data. Generally, the random forest model performed relatively well in predicting both maize foliar temperature and stom-

atal conductance over the various growing stages. Specifically, the random forest model achieved stronger prediction accuracies for foliar temperature than stomatal conductance regarding maize water stress estimation.

The maize foliar temperature was optimally predicted during the mid-vegetative growth stage (RMSE = 0.59 °C, $R^2$ = 0.81 and RRMSE = 2.9%) based on the thermal infrared, followed by the NIR, NGRBI, EVI and NDVI, in order of importance. It has been detailed in the literature that the water content of crop leaves is directly associated with the foliar reflectance across the electromagnetic spectrum [18,72]. Specifically, foliar temperature is strongly detected by the thermal infrared, as it can sense emitted radiant energy; hence, it is commonly used to evaluate crop water stress [27,49,73]. Moreover, the NIR region is valuable when quantifying crop water status due to its strong water absorption ability, which can detect crop water stress-based reflectance variation [74]. Thus, the crop surface temperatures were strongly detected by the thermal infrared and NIR wavebands; hence, they play a crucial role in predicting the maize temperature variability during vegetative growth stages. Correspondingly, the studies of Kullberg, DeJonge [75] and Ma, Xu [76] successfully demonstrated the crucial use of thermal infrared and optical reflectance in the NIR region to retrieve maize water content.

Specifically, during the vegetative stages, the maize canopy structure is under development, exposing the underlying soil surface, which absorbs and retains thermal radiation [23]. The high ambient soil temperatures influence the maize temperature, especially during early vegetative growth, when there is maximum soil exposure. However, as the canopy structure develops, soil influence is reduced, and crops generally experience lower leaf temperatures. Since stomata are generally more numerous on the underside of the leaf, they are influenced by the heat of soil, which reaches the underside due to convection [77,78]. This reduces the stomatal conductance levels, as the crop closes the stomata to retain moisture and subsequently experiences higher foliar temperatures [79]. Hence, the thermal infrared section was a strong predictive variable of foliar temperature due to its ability to overcome the influence of soil temperature during the stages of minimal canopy closure. Leinonen, Grant [80] notes that short-wave solar radiation that is absorbed by the soil affects the vertical canopy surface characteristics.

Nevertheless, the temperature of a leaf, relative to the surrounding air temperature, is primarily influenced by the plant's photosynthetic capacity, as well as the productivity of internal structural leaf components, such as the air cavities, chloroplasts, and mesophyll cell thickness [81,82]. Thus, when a crop is water-stressed, the molecular leaf networks transmit a signal to initiate the physiological and biochemical changes that regularly result in increased foliar temperatures relative to the air temperatures [83–85]. However, when an optimal amount of water is present, the Tc–Ta remains low, as there are productive rates of transpiration and photosynthesis. Therefore, the sustained influence of the thermal infrared- and chlorophyll-based VIs during vegetative growth stages suggested that the crop was optimally transpiring, with no water stress. However, the relatively high maize leaf temperatures, measured in combination with the high Tc–Ta that occurred during the early vegetative stage, suggested slight water stress at crop emergence. In a similar study, Zhang, Niu [23] noted that curled water stressed maize leaves saw an increase in the proportion of sunlit soil and higher temperature; thus, when Tc was extracted from thermal images, the Tc values were higher due to the influence of sunlit soil.

During the reproductive stages, the importance of thermal infrared waveband decreased and spectral wavelengths such as the red-edge and NIR, as well as the VIs derived from these sections, were found to be more important in facilitating the prediction of foliar temperature. This could be attributed to the fact that there was minimal soil exposure due to the fully developed canopy structure. More specifically, the red-edge borders the absorption of photosynthetic pigments such as chlorophyll, which tend to be more vigorous in fully developed canopies [86,87]. Generally, the change in chlorophyll affects the photosynthetic rates, which indirectly alter the temperature tolerance and further the stomatal conductance of the crop, inherently indicating crop water productivity [88]. Additionally,

during these stages, there is a higher leaf area index, facilitating multiple leaf scattering and reducing transmittance through the leaf due to the stronger chlorophyll concentrations that were optimally identified by the NIR region [63,82]. This led to the significant contribution of chlorophyll-based indices from the red-edge and NIR sections, such as the CCCI, NDRE, and GNDVI, during the reproductive stages of temperature prediction. However, hailstorm occurrence during the mid-reproductive stage damaged the maize canopy structure and exposed underlying soil, causing poorer model prediction due to the spectral confusion between soil and foliar temperatures. Subsequent to the hailstorm, the Tc–Ta increased due to the canopy damage, which caused increased crop temperatures. This increase indicated potential crop water stress during the mid-reproductive and late reproductive stages.

Meanwhile, maize stomatal conductance was optimally predicted during the early reproductive stage (RMSE = 25.9 mmol $m^{-2}$ $s^{-1}$, $R^2$ = 0.85 and RRMSE = 11.5%) based on the red-edge, followed by the NIR, green, OSAVI and blue band, in order of importance. Several studies have shown that the red-edge region is renowned for its relationship with plant water stress and evapotranspiration [89–91]. This is because the red-edge is layered with physiological and chemical processes that reflect the photosynthetic activity of the crop, which indicate stomatal conductance and the potential for crop water stress [92,93]. Specifically, during photosynthesis, the red-edge overlaps the fluorescence emission, which affects the magnitude of the reflected spectrum and corroborates stomatal conductance's dependence on photosynthetic activity [91]. Moreover, denser canopies are known to provide increased accuracy in estimates of photosynthetic capacity and stomatal conductance through the NIR region [94,95]. Optimal stomatal activities are also associated with rapid chlorophyll development, which is highly reflected in the red-edge and NIR [89]. In studies by Vitrack-Tamam, Holtzman [91] and Ballester, Brinkhoff [89] the red-edge and NIR bands were noted for their key use in assessing stomatal conductance. This led to the optimal influence of the red-edge and NIR wavebands, as well as the indices derived from these sections to estimate the stomatal conductance of maize in smallholder farms, especially during the reproductive stages.

During the early reproductive stage, the crop is almost at peak biomass and obtains a high leaf surface area, which promotes faster rates of photosynthesis and conductance to support fruit development [19,88]. The measured stomatal activities were most prominent during this stage, as a high foliar surface area is generally associated with the presence of more stomata on the leaf [96]. Hence, transpiration rates are more dynamic, and the leaf stomata open, facilitating high levels of productivity through optimal foliar conductance and the cooling of crop temperatures. Such processes, during the early reproductive stage, indicated crop productivity and an optimal moisture content, which influence the strong reflectance of the leaf tissue in the red-edge and NIR regions. Even though, during these stages, the crop underwent developmental processes that required high amounts of water (tasseling, silking and kernel development), the maize measured an optimal foliar temperature and high stomatal conductance, suggesting crop water productivity and minimal crop water-stress in the smallholder farm. However, the hailstorm, during the mid-reproductive stage disturbed the maize canopy structure and resulted in low measured stomatal conductance. This meant that the thermal infrared became an important predictor due to canopy damage and soil exposure. Furthermore, the high Tc–Ta indicated that hailstorm damage initiated crop water stress, as foliar temperatures increased and stomatal conductance was reduced in the mid-reproductive and later reproductive stages. This was also the case during the early vegetative stage, as stomatal conductance was fairly low and Tc–Ta measurements were high.

Finally, the prediction model proved that a combination of UAV multispectral and thermal wavebands, as well as UAV-derived VIs, can accurately predict maize foliar temperature and stomatal conductance. The variables of importance for both foliar temperature and stomatal conductance were fairly similar in their contribution to model development throughout maize phenotyping. This indicates that foliar temperature and stomatal conductance are independent, yet interrelated functions, which can be holistically used to

understand the potential of crop water stress. Therefore, timeously predicting maize foliar temperature and stomatal conductance allows for smallholder farmers to make near-real-time decisions that aid in water-related crop productivity.

*4.2. Implications of the Study*

Smallholder farming systems often lack the resources to initiate successful farming practices, as commercial agriculture tends to be the focus of contemporary innovation and development. Thus, the findings of this study imply that the incorporation of optical and thermal infrared UAV technology could facilitate an in-depth analysis of near-real-time crop water stress, using temperature and stomatal conductance as proxies. In this regard, the findings of the study are useful to inform smallholder agriculture management by suggesting the potential implementation of irrigation schedules at crucial water stages (i.e., tasseling, silking, and pollination). Moreover, the UAV-derived data identified stages of high temperatures and low stomatal conductance (i.e., early vegetative stage), suggesting a potential moisture deficit and, thus, the need for necessary interventional irrigation schedules to ensure optimal crop productivity and development. Specifically, the irrigation during vegetative stages may aid in optimal productivity and prevent early crop water stress.

The hailstorm during the mid-reproductive stages damaged the maize canopy structure and had adverse effects on crop growth, as well as leading to premature senescence. Subsequent in-field measurements of stomatal conductance reflected potential stress, as stomatal conductance values were much lower than the typical values that were expected during this growth stage. Additionally, foliar temperature measurements were also relatively high and indicated mild stress, especially during the late reproductive stages. These agrometeorological effects prove how South African smallholder farmers are susceptible to weather events that have major consequences on crop water productivity. Thus, the use of UAV-derived data enables the identification of such occurrences in near-real-time, allowing for farmers to make rapid, informed, and effective decisions regarding the subsequent management of the crop. This is crucial, as it also affects the food security and socio-economic growth for smallholder farmers who rely on healthy and moisture-filled crop yields. Therefore, smallholder farmers benefit from the near-real-time analysis of the UAV data, as this can ensure that prompt remedial measures are taken to prevent further crop stress.

*4.3. Limitations and Recommendations for Future Research*

Although the UAV on-board sensor provided a high spatial resolution, the spectral resolution of the sensor limited the multispectral bands and vegetation indices' choices that were derived from these regions. Specifically, a spectral sensor that attains the shortwave infrared section would have been particularly useful to this study, as these wavelengths are instrumental in identifying plant water stress and deriving direct water-related indices. The use of higher spectral resolution data may allow for more precise spectral extraction, especially during the early growth stages, when the crop foliar canopy has not developed. This may have also resulted in an improved model performance and more detailed map outputs for foliar temperature and stomatal conductance across the study area. Therefore, it is recommended that future crop water stress studies explore the use of a camera with additional wavebands, although this needs to be weighed against the additional cost of the camera sensor.

A further limitation of the study was that the training and testing/validation of the RF MLA that was based on a relatively limited record of measured IRT, and stomatal conductance data. These may not adequately represent the variability in these variables, or the climatic conditions experienced within the study area. Furthermore, only two variables were considered during the model development; however, some studies achieved improved success through the incorporation of a third set of independent data that were used to train and test the models. For example, the availability and use of total evapotranspiration

data may have benefitted the study, especially in quantifying potential maize water stress. Thus, it is recommended that a further analysis of model performance is undertaken in the future, when more data become available. Finally, this study specifically examined a maize crop; therefore, it is recommended that research focuses on alternative crops, cultivated by smallholder farmers. Additionally, the consideration of different climatic conditions and different UAV temporal scales could prove to be beneficial.

## 5. Conclusions

Looking at the findings of the study, it can be concluded that foliar temperature and stomatal conductance are adequate indicators to quantify proxies of water stress throughout the growing period. Foliar temperature yielded higher prediction accuracies compared to stomatal conductance. Nevertheless, the random forest regression model optimally predicted both indicators throughout maize phenotyping. Specifically:

- The UAV-derived optical data and thermal infrared waveband optimally estimated maize temperature during the mid-vegetative stage to an RMSE = 0.59 $°C$ and $R^2$ = 0.81 (RRMSE = 2.9%) based on the thermal infrared, followed by the NIR, NGRBI, EVI and NDVI, in order of importance;
- The optical and thermal infrared data optimally predicted stomatal conductance from the early reproductive stage to an RMSE = 25.9 mmol $m^{-2}$ $s^{-1}$ and $R^2$ = 0.85 (RRMSE = 11.5%) based on the red-edge band, followed by the NIR, green, OSAVI and blue band, in order of importance.

Considering the results of the study, UAV technology is a plausible, flexible, and accurate earth observation technique, which is useful for small-scale farming applications. This is because UAV-derived data provide information regarding improved spatial resolutions to help smallholder farmers understand their crop dynamics and make informed farm-management decisions. Specifically, the use of multispectral and thermal infrared UAV data is a step towards the attainment of an agroclimatic, smart, near-real-time and high-spatial-resolution technology to assess crop water stress through foliar temperature and stomatal conductance. However, the study could have benefitted from higher-spectral-resolution data, as well as the measurement of additional variables that could be used for model training and testing to improve model performance. Nevertheless, the random forest model performed relatively well in the estimation of maize leaf temperature and stomatal conductance in the Swayimani area. Therefore, the implementation of low-cost, near-real-time, and evidence-based solutions to smallholder agriculture could prove to be beneficial in facilitating improved interventions in these agricultural systems.

**Author Contributions:** Conceptualisation, K.B., A.C., M.S., V.N., J.O., O.M. and T.M.; methodology, K.B., A.C., M.S., S.G., V.N. and T.M.; software, K.B., V.N., A.C., M.S. and T.M.; validation, K.B., M.S. and A.C.; formal analysis, K.B., M.S. and A.C.; investigation, K.B., M.S., A.C., S.G. and T.M.; resources, K.B., A.C., V.N., V.G.P.C. and T.M.; data curation, K.B.; writing—original draft preparation, K.B.; writing—review and editing, K.B., M.S., A.C., S.G., J.O. and O.M.; supervision, A.C., M.S. and T.M.; project administration, M.S.,T.M. and V.G.P.C.; funding acquisition, T.M. All authors have read and agreed to the published version of the manuscript.

**Funding:** This research was funded by the Water Research Commission of South Africa (WRC) through the Project WRC K5/2971//4 titled the "Use of drones in monitoring crop health, water stress, crop water requirements, and improvements on crop water productivity to enhance precision agriculture and irrigation scheduling", and in part by the National Research Foundation of South Africa (Grant Number: 122140).

**Informed Consent Statement:** Not applicable.

**Data Availability Statement:** The data presented in this study are available on request from the corresponding author. The data are not publicly available due to authorisation restrictions from the funder that limit the distribution of data, as the article is part of an ongoing project where other manuscripts are still being prepared.

**Acknowledgments:** The authors would like to acknowledge the Swayimani smallholder farmers and Swayimani High School. The authors would like to thank Trylee Matongera, Siphiwokuhle Buthelezi, Helen Ndlovu and Amanda Nyawose for their assistance with fieldwork.

**Conflicts of Interest:** The authors declare no conflict of interest.

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
