# Peer review of "Estimation of Maize Foliar Temperature and Stomatal Conductance as Indicators of Water Stress Based on Optical and Thermal Imagery Acquired Using an Unmanned Aerial Vehicle (UAV) Platform"

_drones, doi:10.3390/drones6070169_

Round 1
Reviewer 1 Report
Please find the review attached.

Reviewer 2 Report
The present manuscript is well written and well structured. According to the present reviewer only minor, but very important variations have to be performed before going ahead. In particular, I suggest to consider to talk a little bit of phenological metrics in the introduction section:
please take a look at:
- https://doi.org/10.3390/cli9030047
- https://doi.org/10.2307/3235884
In figure 1 I suggest you top add Datum, Reference system (is it WGS84 ? Please point out it in the maps or caption section) as well as nominal scale, representation scale.
In methods and Material
How do you calibrate the sensor please describe it.
Then, which software do you adopted please better describe the workflow for eg. how do you have performed statistics (in R which package? in Excel ...)
In Figure 11 and figure 12 follow the same advice for figure 1.
Reviewer 3 Report
The content of the manuscript is interesting, but several things need to be improved in the reviewer's opinion
1. in the literature review, the authors referred to papers on similar issues to the subject of their study. These are the articles in the order of their appearance in the references: 13, 23, 25, 34, 38, 39, 40. The Discussion chapter should not only deal with the discussion of the results of their own research, but the authors should refer to at least some of the works cited.
2) The authors did not perform any error analysis of the measurements related to the determination of Variable importance scores of optimal foliar temperature and stomatal conductance bands and VIs throughout the phenological cycle - fig 10. The camera takes several photographs during the drone flight, so there should be some error analysis of the determined indices.
3. the reviewer was not able to find in the text of the manuscript items 83,84,85 which were included in the references section - please indicate where in the text they were referred to.
4. line 206 wrong figure number.
Round 2
Reviewer 3 Report
The authors' answers satisfy me, but:
1. Although the authors of the manuscript corrected the caption of figure 1 on page 4 of the revised manuscript, they did not notice an error in the numbering of the figures on page 7. The figure on page 7 should have been numbered 3 and was erroneously given the number 1. The numbering of the figure should be corrected.
2. I would suggest that the manuscript authors check the MDPI publisher's website for guidance under "Manuscript Preparation" on how to cite authors' names in cited publications and the rules for preparing references.
I believe that these matters in your manuscript need improvement.
